# To beer or not to beer: A meta-analysis of the effects of beer consumption on cardiovascular health

Giorgia Spaggiari[1], Angelo Cignarelli[2]*, Andrea Sansone[3], Matteo Baldi[4], Daniele Santi[1,5]

1 Unit of Endocrinology, Department of Medical Specialties, Azienda Ospedaliero-Universitaria of Modena, Ospedale Civile of Baggiovara, Modena, Italy, 2 Section of Internal Medicine, Endocrinology, Andrology and Metabolic Diseases, Department of Emergency and Organ Transplantation, University of Bari Aldo Moro, Bari, Italy, 3 Center of Reproductive Medicine and Andrology, Institute of Reproductive and Regenerative Biology, Universitätsklinikum Münster, Münster, Germany, 4 Unit of Internal Medicine, Ospedale Civile of Ivrea, Torino, Italy, 5 Unit of Endocrinology, Department of Biomedical, Metabolic and Neural Sciences, University of Modena and Reggio Emilia, Modena, Italy

* angelo.cignarelli@gmail.com

**Data Availability Statement:** All relevant data are within the manuscript (figures and tables).

**Funding:** The author(s) received no specific funding for this work.

## Abstract

A moderate alcohol consumption is demonstrated to exert a protective action in terms of cardiovascular risk. Although this property seems not to be beverage-specific, the various composition of alcoholic compounds could mediate peculiar effects *in vivo*. The aim of this study was to evaluate potential beer-mediated effects on the cardiovascular health in humans, using a meta-analytic approach (trial registration number: CRD42018118387). The literature search, comprising all English articles published until November, 30th 2019 in EMBASE, PubMed and Cochrane database included all controlled clinical trials evaluating the cardiovascular effects of beer assumption compared to alcohol-free beer, water, abstinence or placebo. Both sexes and all beer preparations were considered eligible. Outcome parameters were those entering in the cardiovascular risk charts and those related to endothelial dysfunction. Twenty-six trials were included in the analysis. Total cholesterol was significantly higher in beer drinkers compared to controls (14 studies, 3.52 mg/dL, 1.71–5.32 mg/dL). Similar increased levels were observed in high-density lipoprotein (HDL) cholesterol (18 studies, 3.63 mg/dL, 2.00–5.26 mg/dL) and in apolipoprotein A1 (5 studies, 0.16 mg/dL, 0.11–0.21 mg/dL), while no differences were detected in low density lipoprotein (LDL) cholesterol (12 studies, -2.85 mg/dL, -5.96–0.26 mg/dL) and triglycerides (14 studies, 0.40 mg/dL, -5.00–5.80 mg/dL) levels. Flow mediated dilation (FMD) resulted significantly higher in beer-consumers compared to controls (4 studies, 0.65%, 0.07–1.23%), while blood pressure and other biochemical markers of inflammation did not differ. In conclusion, the specific beer effect on human cardiovascular health was meta-analysed for the first time, highlighting an improvement of the vascular elasticity, detected by the increase of FMD (after acute intake), and of the lipid profile with a significant increase of HDL and apolipoprotein A1 serum levels. Although the long-term effects of beer consumption are not still understood, a beneficial effect of beer on endothelial function should be supposed.

**Competing interests:** The authors have declared that no competing interests exist.

## Introduction

Alcohol consumption is documented to exert multiple consequences on human health. Alongside the well-known harmful effects of alcohol abuse, the real impact of moderate assumption is more complex to elucidate. Several epidemiological studies described a J-shaped correlation between alcohol intake and cardiovascular disease, depicting a dose-related combination of beneficial and harmful effects [1]. Moderate drinkers show lower cardiovascular risk compared to both heavy drinkers and abstainers [2–4]. Indeed, a moderate alcohol consumption, defined as up to 1 drink (12 g of ethanol) daily for women and up to 2 for men, seems to have beneficial effects on general health. In this context, an alcohol-mediated protective action is largely proposed in the cardiovascular setting and numerous underlying mechanisms have been advocated, such as high-density lipoprotein (HDL) increase, low-density lipoprotein (LDL) decrease, reduction in platelet aggregation, beneficial effects on inflammation, anti-atherogenesis and anti-thrombotic actions [5]. Although the qualitative difference between the various alcoholic beverages does not seem to be decisive in reducing cardiovascular risk [6, 7], the *in vivo* effects of wine, beer or spirit assumption result significantly different. Positive effects of moderate alcohol consumption on mortality rate have been reported since the late seventies [8] and have later been reviewed and confirmed by several other papers [9–11]. Indeed, recent meta-analyses demonstrated an overall beneficial effect of both wine and beer consumption on cardiovascular risk and mortality, hypothesizing a fundamental action of the polyphenol content, which is absent in spirits [2, 12–14]. *In vitro* studies suggest a wine-related modulation in the expression of serum inflammatory and leucocyte adhesion molecules, such as the increase of interleukin (IL)-10 and decrease of IL-1α, IL-6, intracellular adhesion molecule (ICAM)-1, P-Selectin, monocyte chemoattractant protein (MCP)-1, and vascular cell-adhesion molecule (VCAM)-1 [12]. While the molecular mechanisms of wine action have been extensively studied, much less is known about beer effects, despite its widespread consumption all over the world [14].

Beer is a product of cereals fermentation and it is composed for 90% by water and for the remaining 10% by sugars, minerals, vitamins, amino acids, alcohol and polyphenols. Alongside the barley and hops-derived polyphenols, other beer ingredients added to impart additional flavours contribute to the polyphenolic content. In particular, the flavonoids depicted in beer are flavan-3-ols (catechin, epicatechin and gallocatechin), flavanols (kaempferol, myricetin and quercetin), and proanthocyanins. Among these, xanthohumol is a prenylated chalcone-type flavonoid representing a beer-specific polyphenol, since it is present exclusively in hop-derived products [15]. This molecule is suggested to confer beer-specific beneficial effects compared to other fermented alcoholic beverages, but this aspect needs further confirmation [13]. Yeasts are also used in the production of beer, as a means to make the drink alcoholic and carbonated. These organisms also produce several compounds having protective effect against degenerative diseases, such as melatonin, tryptophol and serotonin [16] which have also shown beneficial properties in cardiovascular prevention [17–20]. Regarding the alcoholic component, different kinds of beer are available with an alcohol content ranging from 3.5 to 10 percentage weight/volume (% w/v). A moderate beer consumption is defined by the World Health Organization (WHO) as up to a can of 330 mL of beer containing about 5% w/v alcohol daily for women and up to 2 for men [21]. The heterogeneous composition of beer represents one of the most interesting challenge in evaluating effects of alcoholic beverages on human health. Indeed, a hypothetical beneficial effect could be due to the action of either alcohol or polyphenols alone or a synergic effect of both.

Despite the large amount of data investigating the effect of alcoholic beverages on human health, a specific evaluation of the beer potential benefits is not available. Thus, the aim of this

study was to provide a comprehensive evaluation of potential beer-mediated protective effects, focusing on the endothelial health.

## Materials and methods

### Protocol

We performed this study according to the Cochrane Collaboration [22] and PRISMA statement.

### Data sources and strategies

A comprehensive literature search was performed using the following search key words: beer, cardiovascular risk, cardiovascular disease, cardiovascular, endothelial function, endothelial dysfunction, and the Boolean operators AND and OR. All articles published in English language until November, 30th 2019 in EMBASE, PubMed and Cochrane databases were evaluated. The meta-analysis was first registered in the International Prospective Register of Systematic Reviews (PROSPERO registration number: CRD42018118387). Articles were screened to exclude duplicates and trials not aimed at studying beer effects.

### Eligibility criteria

The following inclusion criteria were considered: (i) controlled study design; (ii) adult patients (older than 18 years); (iii) beer administration compared to alcohol-free beer, water, abstinence or placebo. Both sexes were considered eligible, as well as all beer preparations.

### Data process and quality

Two authors extracted data from all randomized controlled trials (RCTs) selected with regard to study design, year of publication, number of included/excluded subjects, inclusion and exclusion criteria and type of intervention. Data about cardiovascular health were extracted from included studies. Since several parameters have been proposed in the literature to evaluate the cardiovascular health, a work diagram was created to highlight those parameters reported in the vast majority of extracted studies. The primary endpoints considered in the analysis were lipid parameters.

The quality of RCTs was assessed using the Cochrane Risk of bias tool.

The procedures followed were in accordance with the ethical standards of the responsible institutional or regional committee on human experimentation or in accordance with the Helsinki Declaration of 1975 as revised in 2013. Considering the meta-analytic study design, ethics approval was not required.

### Data synthesis and analysis

The available data were meta-analyzed using the Review Manager 5.2 software (Version 5.2.4 Copenhagen: The Nordic Cochrane Centre, The Cochrane Collaboration, 2012), using the random or fixed effect models, according to data heterogeneity. Heterogeneity between the results of different studies was examined by inspecting the scatter in the data points and the overlap in their confidence intervals and by performing chi-square tests and $I^2$ statistics. When $I^2$ was higher than 60%, random effect model was used since it provides a more conservative estimate of the overall effect, especially relevant when studies were of different design and duration. Weighted mean differences and 95% confidence intervals were estimated for each endpoint considered. The mean difference was calculated for each study included, using

results reported as mean and standard deviation. When results were reported as median and standard error, these were used to calculate appropriate results.

Publication biases were evaluated inspecting the funnel plot, according to Cochrane collaboration and PRISMA statement. Moreover, the Egger's test was performed to exclude the presence of publication biases.

Sensitivity analyses were performed in order to reduce studies heterogeneity, considering the duration of beer administration in each study enrolled. Indeed, the acute beer administration was separated from the chronic administration. The acute or chronic administration was defined according to the amount of beer administered. In particular, the acute protocol provided a single beer assumption, followed by outcomes measurement. On the contrary, the chronic protocol consisted of many beer administrations in a prolonged time-frame interval and followed by the outcome detection. Sensitivity analyses were performed considering the gender and the health status of the subjects for the significant parameters.

In order to evaluate the alcohol contribution to the potential beneficial effect, meta-regression analyses were performed considering the amount of alcohol used in each trial. Meta-regression analyses were performed using RStudio Server Open Source Edit Version 0.99.902 2016 and R programming software.

Statistical significance was set at $p < 0.05$.

## Results

The literature search detected 2,526,967 studies. After duplicates removal and fitting the search on clinical trials in humans, the final number of articles was 131,495. Among these studies, 69 papers were selected after title and abstract evaluation. Table 1 reported included studies characteristics and S1 Table showed excluded studies and reasons for exclusion. Twenty-six RCTs fully fulfilled inclusion and exclusion criteria and were used for data extraction [11, 14, 23–46] (Fig 1). Among studies enrolled, five trials considered an acute beer administration.

The analysis was limited to those parameters entering in the cardiovascular risk charts and those related to endothelial dysfunction. In general, the most reported parameters were: cholesterol, blood pressure, tumor necrosis factor (TNF), IL-6, flow mediated dilation (FMD), fibrinogen, body mass index (BMI), glucose, insulin, adiponectin and high sensitivity C-reactive protein (hs-CRP). However, not all parameters were available in all studies. Thus, we performed a work diagram (S1 Fig) in which the font dimension is directly related to the number of studies reporting each parameter. Considering those parameters reported in the vast majority of studies (S1 Fig), the analysis was limited to lipid profile, blood pressure, FMD, fibrinogen, adiponectin, hs-CRP, TNF and IL-6.

Total cholesterol was reported in 14 studies, comparing 548 beer drinkers and 532 controls. Total cholesterol was significantly higher in study compared to control group (mean difference 3.52 mg/dL, 1.71 to 5.32 mg/dL, $p<0.001$, $I^2 = 65\%$) (Fig 2). Similar results were found when beer was compared to either alcohol-free beer (mean difference 3.00 mg/dL, 0.51 to 5.49 mg/dL, $p = 0.020$, $I^2 = 69\%$), or placebo/water (mean difference 3.92 mg/dL, 0.51 to 7.33 mg/dL, $p = 0.020$, $I^2 = 60\%$) (Fig 2). Sensitivity analysis was applied to reduce the source of heterogeneity. In studies evaluating only chronic administration of beer, the difference between study and control group was evident comparing beer to alcohol-free beer and not to placebo/water (mean difference 1.92 mg/dL, -4.35 to 8.19 mg/dL, $p = 0.550$, $I^2 = 65\%$). Moreover, no differences were observed dividing studies neither by patients' gender ($p = 0.479$) nor by health status ($p = 0.301$). Meta-regression analyses showed no relationship between total cholesterol and alcohol content ($p = 0.652$). Finally, no publication biases were found ($p = 0.147$).

**Table 1. Included studies characteristics.**

| Author | Journal | Intervention | n | Age (years) | Gender | Inclusion criteria | Aim of the study | Alcohol content | Duration (weeks) | Endpoints evaluated |
|---|---|---|---|---|---|---|---|---|---|---|
| Padro T [14] | Nutrients, 2018 | Beer, alcoholic and not-alcoholic | 36 | 48.3 ±5.4 | Both (21 M, 15 F) | Overweight | Weight, lipoproteins and vascular endothelial function | M: 10 g daily / F: 5 g daily | 4 | cholesterol, hs-CRP, TNF, IL-6, blood pressure, glucose, ASAT, GGT, weight, BMI, waist circumference |
| Tomita J [40] | Biochem Biophys Res Commun. 2017 | Isomerized hop extract | 23 | 28.5±3 | Male | Healthy | Endothelial functions in smokers and non-smokers | NA | acute | FMD |
| Schrieks IC [39] | Alcohol 2016 | Beer, alcoholic and not-alcoholic | 24 | 30.5 ±16.6 | Male | Healthy | Mental stressor attenuates the stress response | 26 g | acute | Stroop task, Trier Social Stress Test, ACTH, Cortisol, DHEA, Interleukins, TNF |
| Morimoto-Kobayashi Y [33] | Nutr J. 2016 | Matured hop bitter acids | 200 | 44.5 ±1.2 | Both (100 M, 100 F) | Overweight | Abdominal body fat reduction | 35 mg daily | 12 | Body weight, visceral fat, BMI, waist and hip circumference, blood pressure, cholesterol, glucose |
| Chiva-Blanch G [26] | Nutr Metab Cardiovasc Dis. 2015 | Beer, alcoholic and not-alcoholic | 33 | 61±6 | Male | High cardiovascular risk | Cardiovascular risk | 30 g daily | 4 | Apolipoprotein A1, adiponectin, cholesterol, Interleukins, MCP-1, VCAM-1, TNF, weight, BMI, waist and hip circumference |
| Karatzi K [32] | Nutrition 2013 | Beer, alcoholic and not-alcoholic | 17 | 28.5 ±5.2 | Male | Healthy | Cardiovascular risk | 6 g | acute | Blood pressure, FMD, Pulse wave velocity |
| Joosten MM [31] | Metabolism. 2011 | Beer, alcoholic and not-alcoholic | 24 | 23.9 ±4.3 | Female | Healthy | Adiponectin | 26 g daily | 3 | Adiponectin, glucose, insulin, cholesterol |
| Imhof A [30] | Diabetes Care. 2009 | Wine, Beer, Vodka | 72 | Range 22–56 | Both (36 M, 36 F) | Healthy | Diabetes risk and cardiovascular mortality | M: 30 g daily / F: 20 g daily | 3 | Adiponectin |
| Imhof A [29] | Diab Vasc Dis Res. 2008 | Wine, Beer, Vodka | 49 | Range 22–56 | Both (25 M, 24 F) | Healthy | Monocyte migration | M: 30 g / F: 20 g | acute | MCP-1,TNF, cholesterol |
| Tousoulis D [41] | Clin Nutr. 2008 | Beer | 83 | 25.1±2 | Male | Healthy | Endothelial function, inflammatory process and thrombosis/fibrinolysis system | 30 g | acute | Glucose, Interleukin-6, cholesterol, TNF, PAI-1, tPA |
| Beulens JW [25] | Nutr Metab Cardiovasc Dis. 2008 | Beer, alcoholic and not-alcoholic | 20 | 20±2 | Male | Healthy | Lipoprotein-associated phospholipase A2 activity | 40 g daily | 3 | Phospholipase A2 activity, cholesterol, BMI, hs-CRP, blood pressure |
| Romeo J [37] | Nutr Metab Cardiovasc Dis. 2008 | Beer | 57 | M: 35 ±6.1 / F: 37.6 ±9.2 | Both (30 M, 27 F) | Healthy | Blood lipid profile | M: 22 g daily / F: 11 g daily | 4 | Waist and hip circumferences, cholesterol, glucose |
| Romeo J [38] | Ann Nutr Metab. 2007 | Beer | 57 | M: 35 ±6.1 / F: 37.6 ±9.2 | Both (30 M, 27 F) | Healthy | Immune function | M: 22 g daily / F: 11 g daily | 4 | Interleukins, IFN, TNF |

(*Continued*)

**Table 1.** (Continued)

| Author | Journal | Intervention | n | Age (years) | Gender | Inclusion criteria | Aim of the study | Alcohol content | Duration (weeks) | Endpoints evaluated |
|---|---|---|---|---|---|---|---|---|---|---|
| Addolorato G [23] | Appetite 2008 | Wine, Beer, Spirits | 40 | 28±6 | Male | Healthy | Oxidative stress and nutritional parameters | 40 g daily | 4 | Malondyaldeide, adenosine-triphosphate, reduced-glutathione, fat mass, cholesterol |
| Beulens JW [24] | Eur J Clin Nutr. 2008 | Beer, alcoholic and not-alcoholic | 20 | 20±2 | Male | Healthy | Adipokines and insulin sensitivity | 40 g daily | 3 | Acylation-stimulating protein, adiponectin, glucose, insulin, leptin |
| Zilkens RR [44] | Hypertension. 2005 | Wine, beer | 24 | 53.3 ±7.7 | Male | Healthy | Vascular function | 41 g daily | 4 | Blood pressure, FMD, glyceryl trinitated-mediated |
| Zilkens RR [45] | J Hypertens. 2003 | Beer | 16 | 51 ±19.8 | Male | Healthy | Conduit artery endothelial function in moderate-to-heavy drinkers (40–110 g/day) | 9.8 g daily | 4 | FMD, GGT, cholesterol, homocysteine blood pressure |
| Sierksma A [11] | Eur J Clin Nutr. 2002 | Beer | 20 | M: 55 ±6 F: 57±4 | Both (10 M, 10 F) | Healthy | The acute phase proteins C-reactive protein and fibrinogen | M: 40 g daily F: 30 g daily | 6 | plasma CRP, fibrinogen, HDL, triglycerides, ALAT, ASAT, GGT |
| van der Gaag MS [43] | J Lipid Res. 2001 | Beer, Wine, Spirits | 11 | Range 45–60 | Male | Healthy | The first two steps of the reverse cholesterol pathway | 40 g daily | 3 | Apolipoprotein A1, cholesterol, paraoxonase |
| van der Gaag MS [42] | Atherosclerosis. 1999 | Beer, Wine, Spirits | 11 | 51.7 ±5.4 | Male | Healthy | Paraoxonase activity in serum | 40 g daily | 3 | paraoxonase activity, apolipoprotein A1, cholesterol |
| Dimmitt SB [27] | Blood Coagul Fibrinolysis. 1998 | Beer | 55 | Range: 20–63 | Male | Healthy | Incidence of ischaemic events | 7.8 g daily | 4 | fibrinogen, PAI-1, tPA |
| Gorinstein S [28] | J Intern Med. 1997 | Beer | 48 | Range: 51–72 | Male | cardiovascular risk | Lipid metabolism and antioxidant activity | 25 g daily | 4 | cholesterol, tocopherol |
| Puddey IB [36] | Hypertension. 1992 | Beer, alcoholic and not-alcoholic | 86 | Range: 25–70 | Male | Overweight | Blood pressure and blood lipids | 3.2 g daily | 18 | weight, blood pressure |
| Puddey IB [34] | Lancet. 1987 | Low-alcoholic beer | 44 | 52.9 ±2.4 | Male | Hypertension | Blood pressure in hypertensive men | 6.4 g daily | 12 | blood pressure |
| Masarei JR [46] | Atherosclerosis. 1986 | Beer | 48 | Range: 25–55 | Male | Healthy | Blood lipid profile | 6.4 g daily | 6 | cholesterol, apolipoprotein-A1 |
| Puddey IB [35] | Hypertension. 1985 | Beer | 46 | Range: 25–55 | Male | Healthy | Blood pressure | 6.4 g daily | 6 | blood pressure |

[ACTH = adrenocorticotropic hormone; ALAT = alanine aminotransferase; ASAT = aspartate aminotransferase; BMI = body mass index; DHEAS = dehydroepiandrosterone sulfate; F = female; FMD = flow mediated dilation; GGT = gamma-glutamyltransferase; HDL = high density lipoprotein; hs-CRP = high sensitivity C-reactive protein; IFN = interferon; IL-6 = interleukin 6; M = male; MMP3 = matrix metalloproteinase-3; MMP9 = matrix metalloproteinase-9; MCP-1 = monocyte chemoattractant protein-1; NA: not available; PAI-1 = plasminogen activator inhibitor-1; TMP1 = tropomyosin 1; tPA = tissue plasminogen activator; TNF = tumor necrosis factor; VCAM-1 = vascular cell adhesion protein 1].

HDL cholesterol was reported in 18 studies, comparing 626 beer drinkers and 635 controls. HDL cholesterol significantly improved in study compared to control group (mean difference 3.63 mg/dL, 2.00 to 5.26 mg/dL, p<0.001, $I^2$ = 96%) (Fig 3). Similar results were found when beer was compared to either alcohol-free beer (mean difference 4.51 mg/dL, 2.23 to 6.79 mg/dL, p<0.001, $I^2$ = 96%), or placebo/water (mean difference 2.59 mg/dL, 0.74 to 4.45 mg/dL, p<0.001, $I^2$ = 93%) (Fig 3). In particular, significant higher apolipoprotein A1 levels were

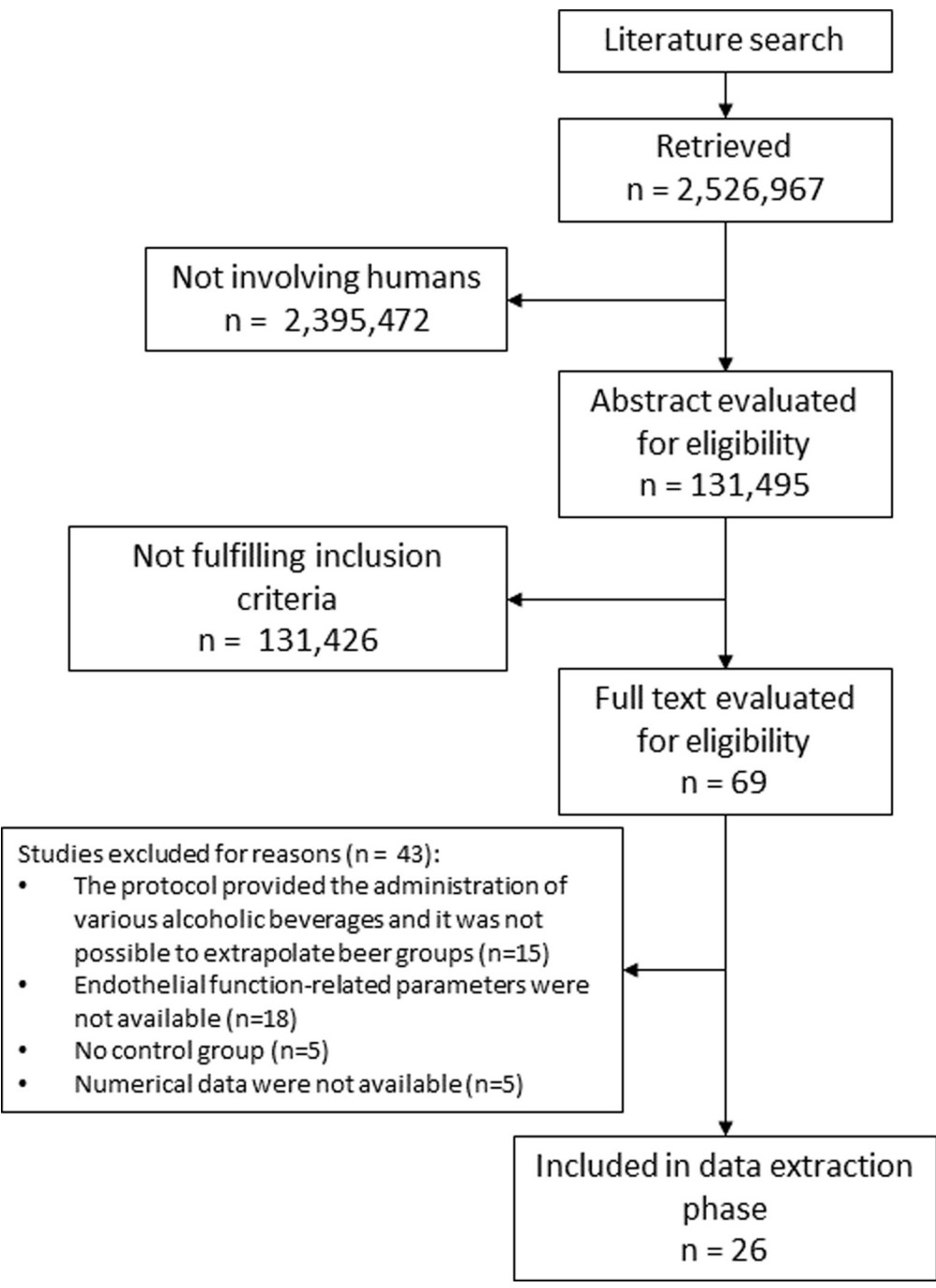

**Fig 1. Flow diagram showing the literature search process.**

detected in study group, compared to control group (mean difference 0.16 mg/dL, 0.11 to 0.21 mg/dL, p<0.001, $I^2$ = 97%) (Fig 4). Sensitivity analysis confirmed these differences also considering exclusively studies assessing chronic administration of beer (mean difference 3.69 mg/dL, 1.74 to 5.63 mg/dL, p<0.001, $I^2$ = 92%). Similar results were obtained dividing studies according to gender (p = 0.709), whereas, the significant change in HDL cholesterol was lost in

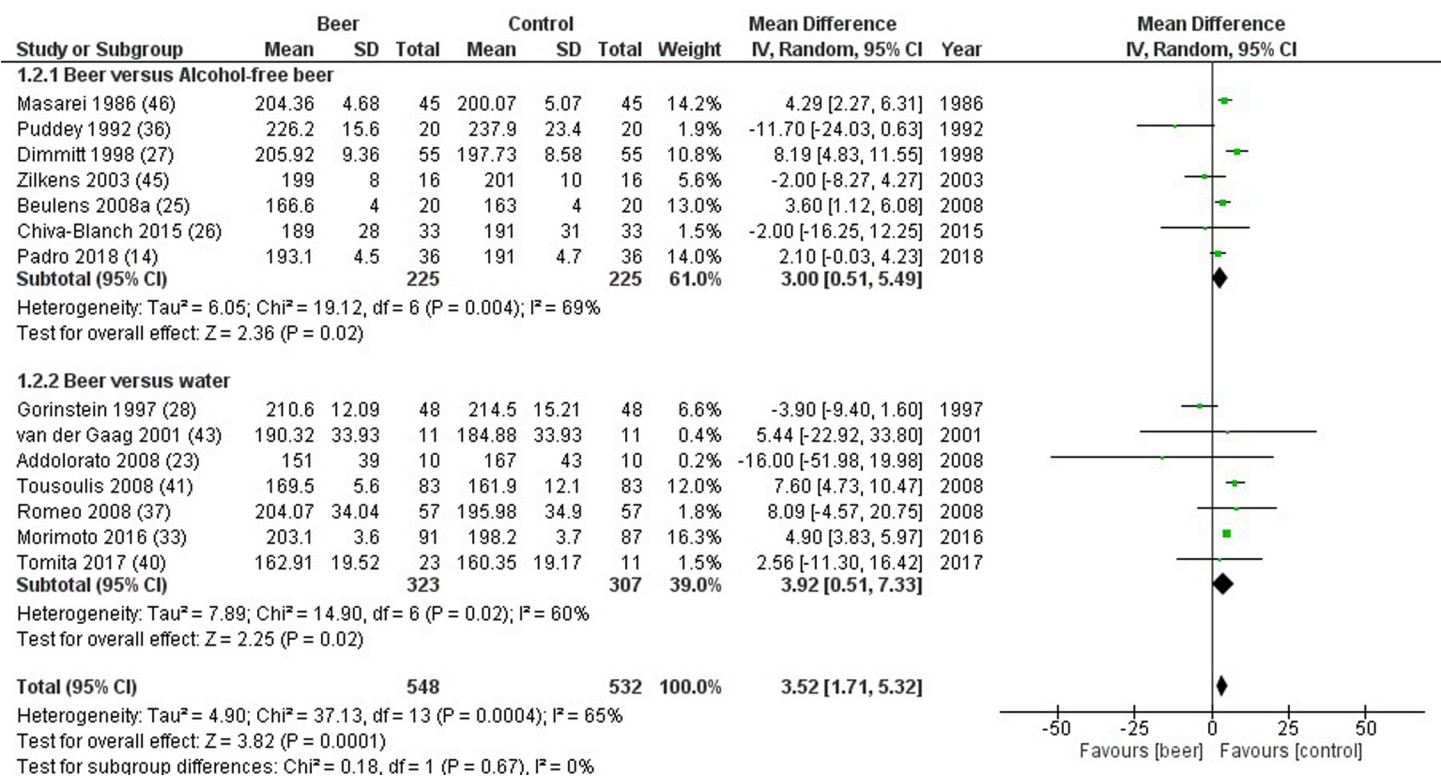

**Fig 2. Forest plot reporting total cholesterol serum levels comparing study and control groups.**

non-healthy subjects (p<0.001). No relationship was found between HDL cholesterol and alcohol content (p = 0.798). Finally, no publication biases were found (p = 0.485).

LDL cholesterol was reported in 12 studies, comparing 459 beer drinkers and 635 controls (S2 Fig). LDL cholesterol was not significantly different between study and control groups (mean difference -2.85 mg/dL, -5.96 to 0.26 mg/dL, p = 0.070, $I^2$ = 95%) (S2 Fig). This lack of difference was maintained also dividing studies considering alcohol-free beer (mean difference -3.13 mg/dL, -6.73 to 0.47 mg/dL, p = 0.090, $I^2$ = 94%), or placebo/water (mean difference -2.36 mg/dL, -7.16 to 2.44 mg/dL, p = 0.330, $I^2$ = 80%) (S2 Fig).

Triglycerides were reported in 14 studies, comparing 552 beer drinkers and 536 controls (S3 Fig). Triglycerides serum levels were not significantly different between study and control groups (mean difference 0.40 mg/dL, -5.00 to 5.80 mg/dL, p = 0.089, $I^2$ = 91%) (S3 Fig). This lack of difference was maintained also dividing studies considering alcohol-free beer (mean difference 2.16 mg/dL, -7.89 to 12.22 mg/dL, p = 0.670, $I^2$ = 96%), or placebo/water (mean difference -0.11 mg/dL, -1.94 to 1.71 mg/dL, p = 0.330, $I^2$ = 0%) (S3 Fig). Sensitivity analysis showed confirmed the lack of significant difference (mean difference 0.54 mg/dL, -1.80 to 2.87 mg/dL, p = 0.720, $I^2$ = 92%).

Blood pressure was evaluated in 8 studies, comparing 363 to 347 subjects. Systolic blood pressure did not differ between study and control groups (mean difference 0.74 mmHg, -0.76 to 2.24 mmHg, p = 0.580, $I^2$ = 97%) (S4 Fig). Similar lack of difference was maintained considering alcohol-free beer (mean difference 1.01 mmHg, -1.41 to 3.43 mmHg, p = 0.410, $I^2$ = 96%) and water/placebo (mean difference 0.07 mmHg, -2.30 to 2.43 mmHg, p = 0.960, $I^2$ = 98%) (S4 Fig). Similarly, diastolic blood pressure did not differ between study and control groups (mean difference 0.15 mmHg, -1.07 to 1.38 mmHg, p = 0.800, $I^2$ = 98%), also

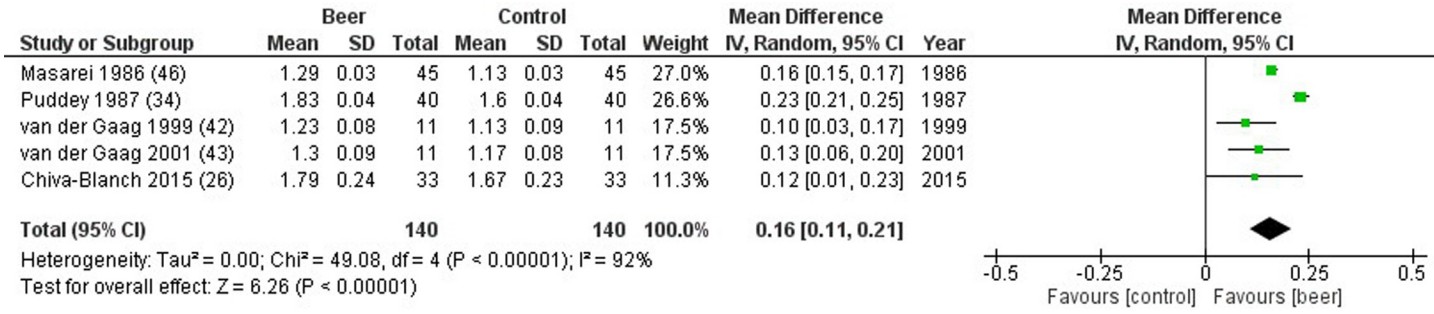

Fig 3. Forest plot reporting high density lipoprotein (HDL) cholesterol serum levels comparing study and control groups.

considering alcohol-free beer (mean difference 1.46 mmHg, -0.02 to 2.94 mmHg, p = 0.050, $I^2$ = 94%) and water/placebo (mean difference -1.73 mmHg, -3.47 to 0.00 mmHg, p = 0.050, $I^2$ = 98%) (S5 Fig). Sensitivity analysis showed confirmed the lack of significant difference (mean difference 0.83 mmHg, -0.71 to 2.37 mmHg, p = 0.290, $I^2$ = 97%).

FMD was reported in 4 studies, resulting significantly higher in the study group compared to controls (mean difference 0.65%, 0.07 to 1.23%, p = 0.030, $I^2$ = 85%) (Fig 5). Considering other biochemical markers of inflammation, no differences were detected for TNF (5 studies,

Fig 4. Forest plot reporting apolipoprotein A1 serum levels comparing study and control groups.

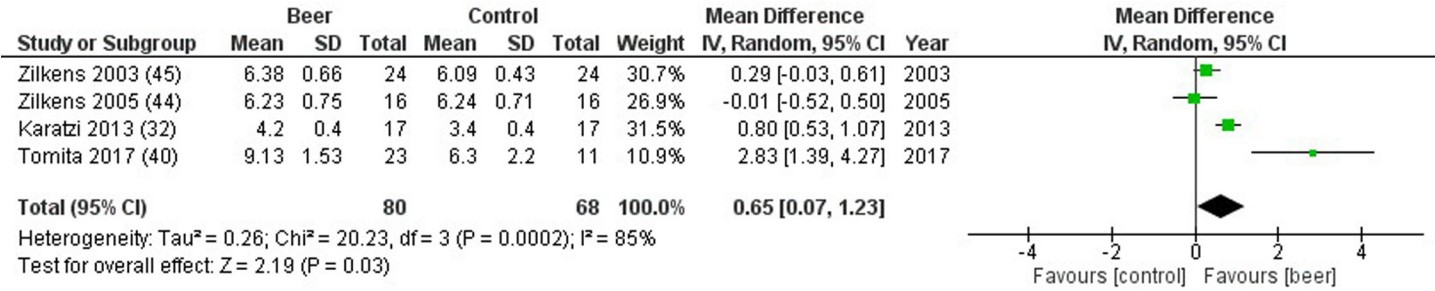

**Fig 5. Forest plot reporting flow mediated dilation (FMD) comparing study and control groups.**

mean difference 0.00 ng/mL, -0.14 to 0.13 ng/mL, p = 0.960, $I^2$ = 85%) (S6 Fig), IL-6 (5 studies, mean difference -1.89 ng/mL, -3.97 to 0.19 ng/mL, p = 0.080, I2 = 91%) (S7 Fig), adiponectin (4 studies, mean difference 0.22 microg/mL, -0.21 to 0.64 microg/mL, p = 0.310, $I^2$ = 93%) (S8 Fig), and hs-CRP (4 studies, mean difference -0.07 microg/mL, -0.34 to 0.19 microg/mL, p = 0.600, $I^2$ = 69%) (S9 Fig). Sensitivity analysis was not performed, since the number of studies included is limited.

Publication bias was not evident in this setting (Fig 6). Risk of bias of was reported in S10 Fig, showing an overall good quality of RCTs included in the final analysis, considering selection, attrition and reporting biases. S11 Fig showed the risk of bias judged for each study included in the analysis. Slightly relevant was the risk of performance and detection biases, which reflected the difficulty to perform a real double-blind clinical trial in the setting of dietary influence on the cardiovascular health.

## Discussion

In Europe, the most prevalent alcoholic beverages are beer, wine and spirits in different proportions and beer represents, in several European countries, the most prevalent source of alcohol consumption (>50%). In this meta-analysis we evaluated, for the first time, the specific effects of beer on human health, distinctly from other alcoholic beverages, such as wine and

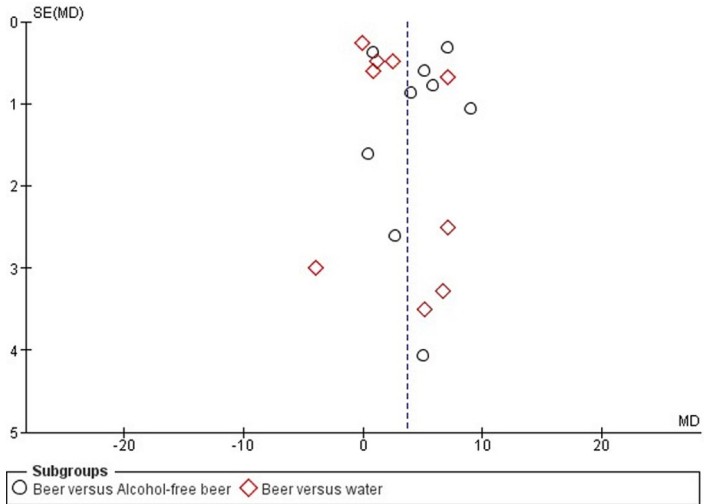

**Fig 6. Tunel plot for high density lipoprotein (HDL) cholesterol levels.**

spirit. Beer consumption is associated with improvement of the vascular elasticity, detected by the increase of FMD and of the lipid profile with a significant increase of HDL cholesterol levels possibly sustained by a raise of apolipoprotein A1 serum levels.

The accumulation of cholesterol within macrophage in arterial wall and their transformation in foam cells represents the key step in the atherosclerosis progression. In this process, HDL has a protective role, promoting the reverse cholesterol transport from peripheral tissues, allowing the excretion of cholesterol out of the body [47]. Thus, considering its physiological role in lipid asset homeostasis, low HDL cholesterol levels represent an independent risk factor for cardiovascular diseases. Here, a relevant effect of beer consumption on cholesterol levels has been demonstrated by an increase of total cholesterol. This increase is supported by a significant raise of HDL level, regardless of a meaningless action on LDL and triglycerides. This is in line with several reports reporting how moderate alcohol consumption may be associated with an increase in either the concentrations or in the anti-oxidant properties of HDL cholesterol [14, 48]. Moreover, the increase of apolipoprotein A1 may represent the mechanism at the base of HDL increase. Padro et al. recently demonstrated that regular consumption of moderate amount (2 cans daily for men and 1 for women) of traditional beer for 4 weeks promoted atheroprotective properties of HDL, preventing LDL oxidation and inducing cholesterol efflux from macrophage [14]. Interestingly, the beneficial effect of beer on HDL cholesterol seems to be evident only in healthy subjects. This specific context should be further analysed by proper-designed trials.

A meta-analytic approach does not allow to discriminate whether lipid modulation is due to alcohol or to polyphenol compounds contained in beer products. Moreover, the alcoholic beverages composition could be extremely heterogeneous, involving a large number of compounds that could play a role in the detected final effect, maybe due to single or synergic mechanisms. As a confirm, meta-regression analyses do not highlight a direct effect of alcohol content on lipid asset, suggesting the action of other molecules. The polyphenols capability to affect the lipid metabolism raising HDL serum levels was suggested by a recent meta-analysis, in which all the possible polyphenol food sources were considered [49]. However, in the present work we considered only beer products, compared to placebo, water, abstinence or alcohol-free beers. Dividing the analysis according to the control group intervention and in particular comparing beer with alcohol-free beer, we aimed to discriminate the alcohol-mediated effects from those due to non-alcoholic components, such as polyphenols. Since the increase in HDL levels remains evident comparing beer assumption to both alcohol-free beer and water, the causal mechanism seems not to be an alcohol-specific competence. However, in quantitative terms, a more consistent HDL increase is observed comparing beer to alcohol-free beer than to water (4.51 *versus* 2.59 mg/dL), suggesting a leading role of alcohol, although not confirmed in meta-regression analyses considering the entire group of studies. Accordingly, previous *in vitro* studies on polyphenols derived from red wine, cocoa and green tea do not show an HDL functionality enhancement [50]. Thus, this beer-related improvement in lipid asset seems to be due mainly to the ethanol itself [15]. This effect could be related to the alcohol dehydrogenase (ADH) activity that, on the other hand, is largely demonstrated to be related to several alcohol-related diseases, such as liver [51] and esophageal squamous cell [52] cancers.

The beer-specific polyphenol xanthohumol exhibits anti-inflammatory properties in *in vitro* studies, inhibiting lipopolysaccharide (LPS)-induced cytokine, MCP-1 and IL-6 production [53]. In addition, the beer polyphenol content is suggested to mediate other cardioprotective actions, such as the increase of endothelial progenitor cells (EPC) and EPC-mobilizing factors, molecules able to repair endothelial damages [54]. However, in our study, we do not highlight a beer-related improvement of TNF, IL-6, adiponectin and hs-CRP. This lack of

polyphenol-mediated improvement in endothelial dysfunction parameters could be simply attributable to the low number of RCTs evaluated or to the presence of other substances able to counteract the polyphenol action [54]. However, an improvement of FMD is detected despite the low number of studies, suggesting a strong effect of beer on endothelial cells and, thus, on vasodilation. FMD is a marker of vascular elasticity generally evaluated at brachial artery at baseline and after 5 minutes of arterial occlusion, showing a normal response to the reactive hyperemia of about 7–15% [55]. Even if the mean increase after beer consumption is only one tenth (0.65%) of the FMD normal value, the significance achieved despite the few included studies suggests a consistent beer-mediated capability to induce vasodilatation. However, this beneficial effect on endothelial function comes mainly from studies in which an acute beer administration is provided, confirming those studies in the literature in which a vascular endothelial function improvement is described after acute beer intake [41]. Indeed, the acute intake could lead to the heart rate increase, the peripheral arteries dilation and the blood pressure reduction.

Beer consumption does not affect blood pressure, neither systolic nor diastolic. This effect is in line with the different properties exhibited by the heterogeneous components of beer. Indeed, the beer-mediated action on blood pressure is due to the vasodilator action of polyphenol [56]. Accordingly, a blood pressure reduction is detected when polyphenols are administered through different oral sources [49], as well as when non-alcoholic polyphenol beer compounds are used [26]. On the other hand, the role of alcohol on blood pressure is still debated. Although the role of excessive alcohol intake on hypertension as well as the blood pressure improvement after alcohol reduction are well established [57], the action of moderate consumption is challenging. However, a possible alcohol-mediated capability to counteract the blood pressure-reducing action of beer polyphenols has already been suggested [54]. Moreover, age, gender and comorbidities of subjects enrolled represent independent factors able to interfere with the blood pressure status evaluation.

An inverse dose-effect correlation between vascular events and alcohol consumption was detected for wine [13, 58] and, more recently, for beer [13]. While a protective range of daily wine consumption (from 10–21 g to 41–72 g) has been proposed, the paucity of available studies do not allow to obtain the same result for beer or spirit [13]. In the studies considered in our meta-analysis, beer is administered in randomized clinical settings in moderate dosages, i.e. less than 330 mL daily. Although the identification of a range of beer safety is beyond the objectives of the present meta-analysis, our results suggest that a moderate beer consumption could exert beneficial effects either in prevention of dyslipidaemia or in vasodilation improvement.

The most reliable outcomes at evaluating the cardiovascular health remains the long-term incidence of major cardiovascular events and the consequent mortality. However, epidemiological studies on alcoholic beverages are limited to the impossibility to precisely assess the type and the amount of alcohol consumed. On the contrary, RCTs allow studying specific types and amounts of alcoholic beverages, although limiting the evaluation on indirect markers of cardiovascular diseases. Despite the clear advantages coming from a controlled clinical setting, RCTs are not free of biases. Indeed, almost all studies included in the analysis did not strictly consider the diet and the activity pattern, except with specific questionnaires [26]. Moreover, the different intervention used in control groups represents another important limit. Indeed, 14 studies (53.8%) compared beer to alcohol-free beers, allowing at detecting mainly the alcohol role, in spite of polyphenolic beer contents. Moreover, the dealcoholisation process provokes a significant loss of non-alcoholic compounds, including polyphenols. Thus, the polyphenol content of beer and alcohol-free beer is not necessarily the same. Finally, since the brand of beers used in RCTs is not always reported, a sub-analysis considering beer styles was not possible.

In conclusion, this is the first comprehensive meta-analysis evaluating beer properties in the cardiovascular setting, suggesting that a moderate beer consumption could beneficially affect HDL serum levels and blood vessels elasticity. However, whether any of our findings has any clinical relevance is a question largely left unanswered, as the small effect sizes do not allow us to reach definite conclusions in that direction. Although the long-term effects of beer consumption are not still understood, a beneficial effect of beer on endothelial function should be supposed.

## Supporting information

**S1 Fig. Work diagram considering all parameters reported in included studies.** The size of the font is derived from the number of times the word is repeated.
(TIF)

**S2 Fig. Forest plot reporting low density lipoprotein (LDL) cholesterol serum levels comparing study and control groups.**
(TIFF)

**S3 Fig. Forest plot reporting triglycerides serum levels comparing study and control groups.**
(TIFF)

**S4 Fig. Forest plot reporting systolic blood pressure comparing study and control groups.**
(TIFF)

**S5 Fig. Forest plot reporting diastolic blood pressure comparing study and control groups.**
(TIFF)

**S6 Fig. Forest plot reporting tumor necrosis factor (TNF) comparing study and control groups.**
(TIFF)

**S7 Fig. Forest plot reporting interleukin-6 (IL-6) comparing study and control groups.**
(TIFF)

**S8 Fig. Forest plot reporting adiponectin comparing study and control groups.**
(TIFF)

**S9 Fig. Forest plot reporting high sensitivity C-reactive protein (hs-CRP) comparing study and control groups.**
(TIFF)

**S10 Fig. Risk of biases evaluated among included studies, using the Cochrane risk of bias tool available at RevMan software.**
(TIF)

**S11 Fig. Risk of biases reported for each trial included in the analysis.** The green dot represents a low risk of bias, the red dot a high risk of bias. The lack of dot represents an intermediate risk of bias. The analysis was performed using the Cochrane risk of bias tool available at RevMan software.
(TIFF)

**S1 Table. Excluded studies characteristics and exclusion reasons.**
(DOCX)

**S1 Checklist. PRISMA 2009 checklist.**
(DOC)

## Author Contributions

**Conceptualization:** Daniele Santi.

**Data curation:** Giorgia Spaggiari, Angelo Cignarelli, Daniele Santi.

**Formal analysis:** Giorgia Spaggiari, Daniele Santi.

**Investigation:** Giorgia Spaggiari, Angelo Cignarelli, Daniele Santi.

**Methodology:** Giorgia Spaggiari, Daniele Santi.

**Project administration:** Daniele Santi.

**Supervision:** Angelo Cignarelli, Andrea Sansone, Daniele Santi.

**Validation:** Angelo Cignarelli, Andrea Sansone, Matteo Baldi, Daniele Santi.

**Visualization:** Matteo Baldi, Daniele Santi.

**Writing – original draft:** Giorgia Spaggiari, Daniele Santi.

**Writing – review & editing:** Angelo Cignarelli, Andrea Sansone, Matteo Baldi.

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
