## [Decision Letter · Decision Letter 0]

10 Mar 2020

PONE-D-20-01229

To beer or not to beer: a meta-analysis of beer consumption on cardiovascular health

PLOS ONE

Dear Dr. Cignarelli,

Thank you for submitting your manuscript to PLOS ONE. After careful consideration, we feel that it has merit but does not fully meet PLOS ONE’s publication criteria as it currently stands. Therefore, we invite you to submit a revised version of the manuscript that addresses the points raised during the review process.

In addition to the comments raised by the two Reviewers, please also consider the following suggestions in your revision: 1. Please supply more information of the trials in Table 1, such as mean age, number of males and females if both sexes were included, the beer administration protocol, such as beer type in terms of percentage weight/volume, and duration of consumption. 2. Would the effect of beer on cardiovascular health differ according to the consumption quantity in terms of gram of ethanol? 3. Figure 2 seemed redundant. Please consider reduce the number of figures by moving some figures from the main manuscript to the supplemental material.

We would appreciate receiving your revised manuscript by Apr 24 2020 11:59PM. To enhance the reproducibility of your results, we recommend that if applicable you deposit your laboratory protocols in protocols.io, where a protocol can be assigned its own identifier (DOI) such that it can be cited independently in the future. For instructions see: http://journals.plos.org/plosone/s/submission-guidelines#loc-laboratory-protocols

We look forward to receiving your revised manuscript.

Kind regards,

Yan Li, MD, PhD

Academic Editor

PLOS ONE

Journal Requirements:

2. Please report the results of the risk assessment individually for each study included. Moreover,   please update the literature search.

Reviewers' comments:

Reviewer's Responses to Questions

**Comments to the Author**

1. Is the manuscript technically sound, and do the data support the conclusions?

Reviewer #1: No

Reviewer #2: Partly

2. Has the statistical analysis been performed appropriately and rigorously? 

Reviewer #1: Yes

Reviewer #2: No

3. Have the authors made all data underlying the findings in their manuscript fully available?

Reviewer #1: No

Reviewer #2: Yes

4. Is the manuscript presented in an intelligible fashion and written in standard English?

Reviewer #1: Yes

Reviewer #2: Yes

5. Review Comments to the Author

Reviewer #1: 1、 In line 84-86, you mentioned “A moderate beer consumption is generally defined as up to a can of 330 mL of beer containing about 4% w/v alcohol daily for women and up to 2 for men.” Is there any related reference to support this definition? According to Ref. “Voskoboinik A, Prabhu S, Ling LH, et al. Alcohol and atrial fibrillation: a sobering review. J Am Coll Cardiol. 2016;68:2567–2576.” , alcohol consumption has been defined as: light (<7 standard drinks/week); moderate (7 to 21 standard drinks/week); and heavy (>21 standard drinks/week), where 1 standard drink is approximately 12 g of alcohol. Furthermore, according to “The World Health Organization’s Estimates of a Standard Drink for Conventional Alcoholic Beverages”（Babor TF, Higgins-Biddle JC. Brief Intervention for Hazardous and Harmful Drinking: A Manual for Use in Primary Care. Geneva, Switzerland: World Health Organization; 2001. http://apps.who.int/iris/bitstream/10665/67210/1/WHO_MSD_MSB_01.6b.pdf)，a standard drink for beer is a can of 330 mL containing about 5% w/v alcohol. Therefore, I think the definition of moderate beer consumption in your manuscript may be not appropriate.

2、 Now that the conclusion of this manuscript is “moderate beer drinkers seem not to be discouraged.” (in line 44-45), Table 1 should include the accurate beer consumption in each study in order to identity whether it is moderate beer consumption or not.

3、 In line 90-92, you mentioned “the aim of this study was to provide a comprehensive evaluation of potential beer-mediated protective effects, focusing on the cardiovascular health”. According to previous studies, cardiovascular health may include weight change, lipid profile, vascular health, blood pressure, inflammation, glycemic control, microbial profile, coagulation and so on. However, why the keywords in your manuscript only included endothelial function? (in line 99-100) And why the primary endpoints considered in the analysis were only lipid parameters? (in line 116) Is this consistent with the study protocol?

4、 In line 135-137, how to define acute beer administration or chronic administration by the duration of beer administration in each study enrolled? And the results of sensitivity analyses should be presented in the main test or supplemental material. As the heterogeneity in your results is pretty high, more sensitivity analyses, such as stratified by age, sex, BMI and so on, should be implemented in order to explore the source of high heterogeneity.

5、 In line 148-152, you mentioned “The most reported parameters were: cholesterol, blood pressure, tumor necrosis factor (TNF), IL-6, flow mediated dilation (FMD), fibrinogen, body mass index (BMI), glucose, insulin, adiponectin and high sensitivity C-reactive protein (hs-CRP). Thus, the analysis was limited to those parameters entering in the cardiovascular risk charts (lipid profile and blood pressure) and those related to endothelial dysfunction (i.e. FMD, fibrinogen, adiponectin, hs-CRP, TNF and IL-6)”. Since BMI, glucose and insulin were also the most reported parameters and were included in the cardiovascular risk charts, why your results did not include these results?

6、 In line 150-152, you said “the analysis was limited to…those related to endothelial dysfunction (i.e. FMD, fibrinogen, adiponectin, hs-CRP, TNF and IL-6)..”. According to previous studies, adiponectin, hs-CRP, TNF and IL-6 should be categoried as inflammation, not as endothelial dysfunction. Fibrinogen should be categoried as coagulation, not as endothelial dysfunction. Please refer to “Haseeb S, Alexander B, Baranchuk A. Wine and Cardiovascular Health: A Comprehensive Review. Circulation. 2017;136(15):1434–1448.”

7、 The retrieval strategies should be more standard and accurate, and should be better list in a table. Please refer to：Lin X, Zhang X, Guo J, et al. Effects of Exercise Training on Cardiorespiratory Fitness and Biomarkers of Cardiometabolic Health: A Systematic Review and Meta-Analysis of Randomized Controlled Trials. J Am Heart Assoc. 2015;4(7):e002014.

8、 Publication bias should be evaluated not only inspecting the Funnel plot, but also implementing the Begg’s adjusted rank correlation test and the Egger’s regression test.

9、 Mean difference in results and figures should be described as “Weighted mean difference (WMD)”

10、 In Figure 1, the reason for excluded records should be specified. And the “Study included in qualitative synthesis” box is redundant.

Reviewer #2: The paper by Spaggiari et al. evaluated the potential effect of beer consumption on cardiovascular health, exemplified as lipid metabolism, endothelial function and blood pressure, using a meta-analytic approach. Totally twenty-six randomized controlled trials were included in the current analyses, beer consumption was associated with higher HDL cholesterol, apoA1 and FMD, whereas not related to blood pressure.

Brief review of the paper:

1) In Abstract, lines 27-28, “…The literature search, comprising all English articles published until November, 30th 2018 in EMBASE…” was inconsistent with the time mentioned in methods (page 5, line 101). Please check.

2) In the last sentence of Abstract, the authors conclude “moderate beer drinkers seem not to be discouraged”. However, Table 1 reporting characteristic of included trials did not include the level of beer consumption. It was hard to conclude whether moderate or high beer drinkers was beneficial from the current analyses. Estimate of the regression coefficient could be used to illustrate the dose-dependent association between beer consumption and cardiovascular markers.

3) Following Question 2, according to eligibility criteria mentioned in Methods (page 5, lines 105-109), only randomized studies were included, but Table 1 emphasized the “randomized trials” in study design which may cause confusion.

4) Of note, wine and spirit-controlled studies were also included, were these active controlled regarded as water? (Figure 4)

5) Please check the data collected in the current analyses carefully. For example, Table 1, the study by Puddey IB (Hypertension 1992) was done in 86 overweight men (not healthy men) for an 18-week intervention of low or normal alcohol intake (not beer, alcoholic and not-alcoholic).[1]

6) The acute effect of beer should be separated from the long-term effect, especially on blood pressure and FMD. Acute beer consumption would increase heart rate, dilate peripheral arteries and therefore reduce blood pressure, whether the continuing decline after long-term of alcohol consumption was of interest.

[1] Puddey IB, Parker M, Beilin LJ, et al. Effects of alcohol and caloric restrictions on blood pressure and serum lipids in overweight men. Hypertension. 1992;20:533-41.

6. PLOS authors have the option to publish the peer review history of their article (what does this mean?). If published, this will include your full peer review and any attached files.

Reviewer #1: No

Reviewer #2: No

---

## [Author Response · Author response to Decision Letter 0]

2 Apr 2020

Editorial assistant comments

1. Please report the results of the risk assessment individually for each study included. Moreover, please update the literature search.

ANSWER: The risk assessment has been performed evaluating the publication biases, using the Eggert’s regression as the reviewers suggested. The other risk of biases providing by Cochrane collaboration could not be performed in this kind of meta-analysis. We better explain this within the text. Moreover, we updated the literature search as reported in the methods section.

ANSWER: Within the ‘data availability’ section in the submission process, we stated that “Since this is a meta-analysis, all relevant data are within the manuscript (figures and tables)”.

Associate Editor

Comment 1

In addition to the comments raised by the two Reviewers, please also consider the following suggestions in your revision: 1. Please supply more information of the trials in Table 1, such as mean age, number of males and females if both sexes were included, the beer administration protocol, such as beer type in terms of percentage weight/volume, and duration of consumption. 

ANSWER: We updated the table 1, including more data for each study included.

Comment 2

Would the effect of beer on cardiovascular health differ according to the consumption quantity in terms of gram of ethanol? 

ANSWER: Thank you for your suggestion. We included a meta-regression analysis within the text (in both materials and methods and Results sections) to evaluate the role of alcohol content on the outcome change. In particular, we focused these further analyses on the main outcomes obtained (i.e. total and HDL cholesterol).

Comment 3

Figure 2 seemed redundant. Please consider reduce the number of figures by moving some figures from the main manuscript to the supplemental material.

ANSWER: We moved the figure 2 within supplementary materials. We changed all figures order within the text.

Reviewer #1: 

Comment 1

In line 84-86, you mentioned “A moderate beer consumption is generally defined as up to a can of 330 mL of beer containing about 4% w/v alcohol daily for women and up to 2 for men.” Is there any related reference to support this definition? According to Ref. “Voskoboinik A, Prabhu S, Ling LH, et al. Alcohol and atrial fibrillation: a sobering review. J Am Coll Cardiol. 2016;68:2567–2576.” , alcohol consumption has been defined as: light (<7 standard drinks/week); moderate (7 to 21 standard drinks/week); and heavy (>21 standard drinks/week), where 1 standard drink is approximately 12 g of alcohol. Furthermore, according to “The World Health Organization’s Estimates of a Standard Drink for Conventional Alcoholic Beverages”（Babor TF, Higgins-Biddle JC. Brief Intervention for Hazardous and Harmful Drinking: A Manual for Use in Primary Care. Geneva, Switzerland: World Health Organization; 2001. http://apps.who.int/iris/bitstream/10665/67210/1/WHO_MSD_MSB_01.6b.pdf)，a standard drink for beer is a can of 330 mL containing about 5% w/v alcohol. Therefore, I think the definition of moderate beer consumption in your manuscript may be not appropriate.

ANSWER: Thank you for your observation. We referred to the WHO definition and we corrected the sentence as follows: “A moderate beer consumption is defined by the World Health Organization (WHO) as up to a can of 330 mL of beer containing about 5% w/v alcohol daily for women and up to 2 for men (21).”, including the reference to Babor et al.

Comment 2

Now that the conclusion of this manuscript is “moderate beer drinkers seem not to be discouraged.” (in line 44-45), Table 1 should include the accurate beer consumption in each study in order to identity whether it is moderate beer consumption or not.

ANSWER: We updated the table 1, including detailed information about beer consumption.

Comment 3 

In line 90-92, you mentioned “the aim of this study was to provide a comprehensive evaluation of potential beer-mediated protective effects, focusing on the cardiovascular health”. According to previous studies, cardiovascular health may include weight change, lipid profile, vascular health, blood pressure, inflammation, glycemic control, microbial profile, coagulation and so on. However, why the keywords in your manuscript only included endothelial function? (in line 99-100) And why the primary endpoints considered in the analysis were only lipid parameters? (in line 116) Is this consistent with the study protocol?

ANSWER: We changed the aim of the study, specifying the more focused endpoint: i.e. endothelial function instead of the most general cardiovascular function. To obtain the largest number of manuscripts on this topic, we enlarged the selection of study during the literature search, using potentially wide key words, such as “cardiovascular health”, endothelial dysfunction” and so on. This search reported 26 trials after the duplicate removal and the selection reported in the materials and methods section. In each study, a very large number of potential endpoints were available. In order to focus the analysis on those endpoints reported in the vast majority of studies included, we performed a first analysis to identify the most significant (in terms of number of availability) outcomes. This result was graphed into Figure 2, now supplementary figure 1. We highlighted this decision process within the text as follows: “The analysis was limited to those parameters entering in the cardiovascular risk charts and those related to endothelial dysfunction. In general, the most reported parameters were: cholesterol, blood pressure, tumor necrosis factor (TNF), IL-6, flow mediated dilation (FMD), fibrinogen, body mass index (BMI), glucose, insulin, adiponectin and high sensitivity C-reactive protein (hs-CRP). However, not all parameters were available in all studies. Thus, we performed a work diagram (Supplementary figure 1) in which the font dimension is directly related to the number of studies reporting each parameter. Considering those parameters reported in the vast majority of studies (Supplementary figure 1), the analysis was limited to lipid profile, blood pressure, FMD, fibrinogen, adiponectin, hs-CRP, TNF and IL-6”.

Comment 4

In line 135-137, how to define acute beer administration or chronic administration by the duration of beer administration in each study enrolled? And the results of sensitivity analyses should be presented in the main test or supplemental material. As the heterogeneity in your results is pretty high, more sensitivity analyses, such as stratified by age, sex, BMI and so on, should be implemented in order to explore the source of high heterogeneity.

ANSWER: Thank you for your suggestion.

The distinction between acute and chronic beer administration depends on the time of outcome measure. In particular, whether the protocol provided one beer administration, followed by outcomes measurement, it was defined as acute. On the contrary, if the outcome measure occurred after many beer administrations, it defined a chronic assumption. We better specified this point within the materials and methods section as follows: “The acute or chronic administration was defined according to the amount of beer administered. In particular, the acute protocol provided a single beer assumption, followed by outcomes measurement. On the contrary, the chronic protocol consisted of many beer administrations in a prolonged time-frame interval and followed by the outcome detection”.

Moreover, we included new sensitivity analyses considering the gender and the health status of the subjects. We included these new analyses in both the materials and methods and in the results sections.

Comment 5、 

In line 148-152, you mentioned “The most reported parameters were: cholesterol, blood pressure, tumor necrosis factor (TNF), IL-6, flow mediated dilation (FMD), fibrinogen, body mass index (BMI), glucose, insulin, adiponectin and high sensitivity C-reactive protein (hs-CRP). Thus, the analysis was limited to those parameters entering in the cardiovascular risk charts (lipid profile and blood pressure) and those related to endothelial dysfunction (i.e. FMD, fibrinogen, adiponectin, hs-CRP, TNF and IL-6)”. Since BMI, glucose and insulin were also the most reported parameters and were included in the cardiovascular risk charts, why your results did not include these results?

ANSWER: We agree with your suggestion. However, we did not consider BMI, glucose and insulin in the analysis, since they were reported in a limited number of studies included. This is reported in the actual supplementary figure 1. We specified this point in the main text as follows: “The analysis was limited to those parameters entering in the cardiovascular risk charts and those related to endothelial dysfunction. In general, the most reported parameters were: cholesterol, blood pressure, tumor necrosis factor (TNF), IL-6, flow mediated dilation (FMD), fibrinogen, body mass index (BMI), glucose, insulin, adiponectin and high sensitivity C-reactive protein (hs-CRP). However, not all parameters were available in all studies. Thus, we performed a work diagram (Supplementary figure 1) in which the font dimension is directly related to the number of studies reporting each parameter. Considering those parameters reported in the vast majority of studies (Supplementary figure 1), the analysis was limited to lipid profile, blood pressure, FMD, fibrinogen, adiponectin, hs-CRP, TNF and IL-6.”

Comment 6

In line 150-152, you said “the analysis was limited to…those related to endothelial dysfunction (i.e. FMD, fibrinogen, adiponectin, hs-CRP, TNF and IL-6)..”. According to previous studies, adiponectin, hs-CRP, TNF and IL-6 should be categoried as inflammation, not as endothelial dysfunction. Fibrinogen should be categoried as coagulation, not as endothelial dysfunction. Please refer to “Haseeb S, Alexander B, Baranchuk A. Wine and Cardiovascular Health: A Comprehensive Review. Circulation. 2017;136(15):1434–1448.”

ANSWER: We changed this sentence, according to your previous comment and in order to avoid confusion and to be more polite. Now, this paragraph is as follows: “The analysis was limited to those parameters entering in the cardiovascular risk charts and those related to endothelial dysfunction. In general, the most reported parameters were: cholesterol, blood pressure, tumor necrosis factor (TNF), IL-6, flow mediated dilation (FMD), fibrinogen, body mass index (BMI), glucose, insulin, adiponectin and high sensitivity C-reactive protein (hs-CRP). However, not all parameters were available in all studies. Thus, we performed a work diagram (Supplementary figure 1) in which the font dimension is directly related to the number of studies reporting each parameter. Considering those parameters reported in the vast majority of studies (Supplementary figure 1), the analysis was limited to lipid profile, blood pressure, FMD, fibrinogen, adiponectin, hs-CRP, TNF and IL-6.”

Comment 7

The retrieval strategies should be more standard and accurate, and should be better list in a table. Please refer to：Lin X, Zhang X, Guo J, et al. Effects of Exercise Training on Cardiorespiratory Fitness and Biomarkers of Cardiometabolic Health: A Systematic Review and Meta-Analysis of Randomized Controlled Trials. J Am Heart Assoc. 2015;4(7):e002014.

ANSWER: Dear reviewer, the description of the literature search could be useful but, at the same time, confusing. We followed the indication proposed by the Cochrane collaboration and we decided to keep the literature search description as suggested by their statement. We included, in the materials and methods section, the reference of Cochrane collaboration handbook (Higgins JPT, Thomas J, Chandler J, Cumpston M, Page MJ, Welch VA. Cochrane Handbook for Systematic Reviews of Interventions 2nd Edition. Higgins JPT, Thomas J, Chandler J, Cumpston M, Page MJ, Welch VA, editors. Chichester (UK): John Wiley & Sons; 2019).

Comment 8 

Publication bias should be evaluated not only inspecting the Funnel plot, but also implementing the Begg’s adjusted rank correlation test and the Egger’s regression test.

ANSWER: Thank you for your suggestion. We included the Eggert’s analysis to evaluate the possible presence of publication biases.

Comment 9 

Mean difference in results and figures should be described as “Weighted mean difference (WMD)”

ANSWER: The outcome measurement is quite standard for those endpoints reported. Thus, as suggested by the Cochrane Collaboration, it is no mandatory to report differences as standardized mean difference, but it should be reported as mean differences.

Comment 10 

In Figure 1, the reason for excluded records should be specified. And the “Study included in qualitative synthesis” box is redundant.

ANSWER: Thank you. We updated the figure 1 as you suggested.

Reviewer #2: 

The paper by Spaggiari et al. evaluated the potential effect of beer consumption on cardiovascular health, exemplified as lipid metabolism, endothelial function and blood pressure, using a meta-analytic approach. Totally twenty-six randomized controlled trials were included in the current analyses, beer consumption was associated with higher HDL cholesterol, apoA1 and FMD, whereas not related to blood pressure.

Brief review of the paper:

Comment 1 

In Abstract, lines 27-28, “…The literature search, comprising all English articles published until November, 30th 2018 in EMBASE…” was inconsistent with the time mentioned in methods (page 5, line 101). Please check.

ANSWER: Thank you, we aligned the two parts

Comment 2

In the last sentence of Abstract, the authors conclude “moderate beer drinkers seem not to be discouraged”. However, Table 1 reporting characteristic of included trials did not include the level of beer consumption. It was hard to conclude whether moderate or high beer drinkers was beneficial from the current analyses. Estimate of the regression coefficient could be used to illustrate the dose-dependent association between beer consumption and cardiovascular markers.

ANSWER: Thank you for this observation. According to your suggestion and to previous comments, we updated the Table 1, reporting all data available and useful to understand the alcohol intake during the study protocol.

Comment 3 

Following Question 2, according to eligibility criteria mentioned in Methods (page 5, lines 105-109), only randomized studies were included, but Table 1 emphasized the “randomized trials” in study design which may cause confusion.

ANSWER: Thank you. We deleted the “randomized trials” in the table 1

Comment 4

Of note, wine and spirit-controlled studies were also included, were these active controlled regarded as water? (Figure 4)

ANSWER: In studies in which other alcohol beverages were used, only the group in which beer was administered were considered and compared to water or alcohol-free beer.

Comment 5

Please check the data collected in the current analyses carefully. For example, Table 1, the study by Puddey IB (Hypertension 1992) was done in 86 overweight men (not healthy men) for an 18-week intervention of low or normal alcohol intake (not beer, alcoholic and not-alcoholic).[1]

ANSWER: We corrected the definition of included patients in the work by Puddey et al. 1992. However, in this trial it is possible to obtain those patients in which only beer was administered.

Comment 6 

The acute effect of beer should be separated from the long-term effect, especially on blood pressure and FMD. Acute beer consumption would increase heart rate, dilate peripheral arteries and therefore reduce blood pressure, whether the continuing decline after long-term of alcohol consumption was of interest.

ANSWER: Thank you for your suggestion. We included this idea within the discussion section as follows: “The beneficial effect of acute beer intake on vascular endothelial function is well described (41), leading to the heart rate increase, the peripheral arteries dilation and the blood pressure reduction. In line with this aspect, our results support that this improvement is maintained also for moderate beer intake, especially considering blood pressure and FMD.”

---

## [Decision Letter · Decision Letter 1]

29 Apr 2020

PONE-D-20-01229R1

To beer or not to beer: a meta-analysis of beer consumption on cardiovascular health

PLOS ONE

Dear Dr. Cignarelli,

Thank you for submitting your manuscript to PLOS ONE. After careful consideration, we feel that it has merit but does not fully meet PLOS ONE’s publication criteria as it currently stands. Therefore, we invite you to submit a revised version of the manuscript that addresses the points raised during the review process.

Specifically, please, do check if the included trials reached the inclusion criteria of the current investigation, and ensure data accuracy of the tables and figures. 

We would appreciate receiving your revised manuscript by Jun 13 2020 11:59PM. To enhance the reproducibility of your results, we recommend that if applicable you deposit your laboratory protocols in protocols.io, where a protocol can be assigned its own identifier (DOI) such that it can be cited independently in the future. For instructions see: http://journals.plos.org/plosone/s/submission-guidelines#loc-laboratory-protocols

We look forward to receiving your revised manuscript.

Kind regards,

Yan Li, MD, PhD

Academic Editor

PLOS ONE

Reviewers' comments:

Reviewer's Responses to Questions

**Comments to the Author**

1. If the authors have adequately addressed your comments raised in a previous round of review and you feel that this manuscript is now acceptable for publication, you may indicate that here to bypass the “Comments to the Author” section, enter your conflict of interest statement in the “Confidential to Editor” section, and submit your "Accept" recommendation.

Reviewer #1: (No Response)

Reviewer #2: (No Response)

2. Is the manuscript technically sound, and do the data support the conclusions?

Reviewer #1: No

Reviewer #2: Partly

3. Has the statistical analysis been performed appropriately and rigorously? 

Reviewer #1: Yes

Reviewer #2: Yes

4. Have the authors made all data underlying the findings in their manuscript fully available?

Reviewer #1: Yes

Reviewer #2: Yes

5. Is the manuscript presented in an intelligible fashion and written in standard English?

Reviewer #1: Yes

Reviewer #2: Yes

6. Review Comments to the Author

Reviewer #1: Thanks for your effort in modifying this manuscript, most of my questions have been solved. But I still have some questions and suggestions as follows:

1. Please re-confirm your data in table 1 precisely!

On the one hand, in line 108 to 111, you mentioned that “The following inclusion criteria were considered: (i) double-blind, randomized, controlled study design”, however, Ref “Zilkens RR, Burke V, Hodgson JM, et al. Red wine and beer elevate blood pressure in normotensive men. Hypertension. 2005;45(5):874–879”,” Padro T, Muñoz-García N, Vilahur G, et al. Moderate Beer Intake and Cardiovascular Health in Overweight Individuals. Nutrients. 2018;10(9):1237”, “Schrieks IC, Joosten MM, Klöpping-Ketelaars WA, Witkamp RF, Hendriks HF. Moderate alcohol consumption after a mental stressor attenuates the endocrine stress response. Alcohol. 2016;57:29–34”, “Chiva-Blanch G, Magraner E, Condines X, et al. Effects of alcohol and polyphenols from beer on atherosclerotic biomarkers in high cardiovascular risk men: a randomized feeding trial. Nutr Metab Cardiovasc Dis. 2015;25:36–45”, Karatzi K, Rontoyanni VG, Protogerou AD, et al. Acute effects of beer on endothelial function and hemodynamics: a single-blind, crossover study in healthy volunteers. Nutrition. 2013;29:1122-6.”, “Joosten MM, Witkamp RF, Hendriks HF. Alterations in total and high-molecular-weight adiponectin after 3 weeks of moderate alcohol consumption in premenopausal women. Metabolism. 2011;60:1058–1063”, “Imhof A, Plamper I, Maier S, et al. Effect of drinking on adiponectin in healthy men and women: a randomized intervention study of water, ethanol, red wine, and beer with or without alcohol. Diabetes Care. 2009;32:1101–1103” and so on are even not double-blind study.

On the other hand, in Ref “Tomita J, Mochizuki S, Fujimoto S, et al. Acute improvement of endothelial functions after oral ingestion of isohumulones, bitter components of beer. Biochem Biophys Res Commun. 2017;484(4):740–745”, the total number is 23, not 31.

In conclusion, the accuracy of the data in this article is questionable.

2. In line 39-40, you mentioned “In conclusion, the specific beer effect on human cardiovascular health was evaluated for the first time”, in line 329-331, you also mentioned “In conclusion, this is the first comprehensive evaluation of beer properties in the cardiovascular setting…”, however, as far as I know, there exists some researches have already evaluated the beer effect on human cardiovascular health, do you mean this paper is “the first comprehensive meta-analysis specific beer effect on human cardiovascular health”? Please clarify.

3. In table 1 and each forest plots, please add the corresponding reference number so as to assist the reader to find the reference.

Reviewer #2: 1) As is clarified in the revised manuscript (page 4 lines 4-6), a moderate beer consumption was defined by the World Health Organization (WHO) as up to a can of 330 mL of beer containing about 5% w/v alcohol daily for women and up to 2 for men, which was equivalent to 16.5g/d for women and 33g/d for men. However, the levels of alcohol content were beyond this range in several studies listed in Table 1. Could they be regarded as moderate beer consumption?

2) When it comes to the beneficial effect of beer on endothelial function raised by authors (page 2, lines 42-44, page 14, lines 286-295), the conclusion was based on the results of 4 trials, of which 2 were acute effect whereas others were chronic effect. The improvement of FMD was predominantly derived from the significant acute effect of two trials (Figure 5), which should be emphasized in conclusion. Following Reviewer 2 Comment 6, the beneficial effect of beer on endothelial function remained to be illustrated.

3) On page 12, lines 246-247, the authors discussed “The effect on total cholesterol seems to be mainly related to the acute effect rather than the chronic one”, which was inconsistent with the results of sensitivity analyses, the difference between beer group and alcohol-free beer group was statistically significant (page 8, lines 172-175).

7. PLOS authors have the option to publish the peer review history of their article (what does this mean?). If published, this will include your full peer review and any attached files.

Reviewer #1: No

Reviewer #2: No

---

## [Author Response · Author response to Decision Letter 1]

2 May 2020

Comments to the Author

Reviewer #1

Thanks for your effort in modifying this manuscript, most of my questions have been solved. But I still have some questions and suggestions as follows:

Comment 1

1. Please re-confirm your data in table 1 precisely!

ANSWER: We revised the table 1.

Comment 2

On the one hand, in line 108 to 111, you mentioned that “The following inclusion criteria were considered: (i) double-blind, randomized, controlled study design”, however, Ref “Zilkens RR, Burke V, Hodgson JM, et al. Red wine and beer elevate blood pressure in normotensive men. Hypertension. 2005;45(5):874–879”,” Padro T, Muñoz-García N, Vilahur G, et al. Moderate Beer Intake and Cardiovascular Health in Overweight Individuals. Nutrients. 2018;10(9):1237”, “Schrieks IC, Joosten MM, Klöpping-Ketelaars WA, Witkamp RF, Hendriks HF. Moderate alcohol consumption after a mental stressor attenuates the endocrine stress response. Alcohol. 2016;57:29–34”, “Chiva-Blanch G, Magraner E, Condines X, et al. Effects of alcohol and polyphenols from beer on atherosclerotic biomarkers in high cardiovascular risk men: a randomized feeding trial. Nutr Metab Cardiovasc Dis. 2015;25:36–45”, Karatzi K, Rontoyanni VG, Protogerou AD, et al. Acute effects of beer on endothelial function and hemodynamics: a single-blind, crossover study in healthy volunteers. Nutrition. 2013;29:1122-6.”, “Joosten MM, Witkamp RF, Hendriks HF. Alterations in total and high-molecular-weight adiponectin after 3 weeks of moderate alcohol consumption in premenopausal women. Metabolism. 2011;60:1058–1063”, “Imhof A, Plamper I, Maier S, et al. Effect of drinking on adiponectin in healthy men and women: a randomized intervention study of water, ethanol, red wine, and beer with or without alcohol. Diabetes Care. 2009;32:1101–1103” and so on are even not double-blind study.

ANSWER: Thank you for your observation. This is a misprint. Indeed, we included all controlled trials, not necessarily double-blinded. We corrected the inclusion criteria part accordingly.

Comment 2

On the other hand, in Ref “Tomita J, Mochizuki S, Fujimoto S, et al. Acute improvement of endothelial functions after oral ingestion of isohumulones, bitter components of beer. Biochem Biophys Res Commun. 2017;484(4):740–745”, the total number is 23, not 31.

In conclusion, the accuracy of the data in this article is questionable.

ANSWER: We revised the manuscript, as reported in comment 1

Comment 3

In line 39-40, you mentioned “In conclusion, the specific beer effect on human cardiovascular health was evaluated for the first time”, in line 329-331, you also mentioned “In conclusion, this is the first comprehensive evaluation of beer properties in the cardiovascular setting…”, however, as far as I know, there exists some researches have already evaluated the beer effect on human cardiovascular health, do you mean this paper is “the first comprehensive meta-analysis specific beer effect on human cardiovascular health”? Please clarify.

ANSWER: Thank you. We agree and we corrected the discussion as you suggested.

Comment 4

In table 1 and each forest plots, please add the corresponding reference number so as to assist the reader to find the reference.

ANSWER: We added the references to table 1 and to forest plots.

Reviewer #2: 

Comment 1

As is clarified in the revised manuscript (page 4 lines 4-6), a moderate beer consumption was defined by the World Health Organization (WHO) as up to a can of 330 mL of beer containing about 5% w/v alcohol daily for women and up to 2 for men, which was equivalent to 16.5g/d for women and 33g/d for men. However, the levels of alcohol content were beyond this range in several studies listed in Table 1. Could they be regarded as moderate beer consumption?

ANSWER: Thank you for this clarification. We considered that an average moderate beer consumption could be considered in our work. Indeed, first only some studies the daily beer administration was higher than those defined as moderate consumption, although this was a slight increase. Second, in these works, the authors themselves defined these dosages as moderate consumption. Thus, considering together these points, we could consider that on average this is a moderate beer consumption.

Comment 2

When it comes to the beneficial effect of beer on endothelial function raised by authors (page 2, lines 42-44, page 14, lines 286-295), the conclusion was based on the results of 4 trials, of which 2 were acute effect whereas others were chronic effect. The improvement of FMD was predominantly derived from the significant acute effect of two trials (Figure 5), which should be emphasized in conclusion. Following Reviewer 2 Comment 6, the beneficial effect of beer on endothelial function remained to be illustrated.

ANSWER: It is true. Probably, the beneficial effect of beer consumption on endothelial function is mainly due to acute administration, rather than chronic. We added this speculation in the discussion section as follows:

“However, this beneficial effect on endothelial function comes mainly from studies in which an acute beer administration is provided, confirming those studies in the literature in which a vascular endothelial function improvement is described after acute beer intake (41). Indeed, the acute intake could lead to the heart rate increase, the peripheral arteries dilation and the blood pressure reduction“.

Comment 3

On page 12, lines 246-247, the authors discussed “The effect on total cholesterol seems to be mainly related to the acute effect rather than the chronic one”, which was inconsistent with the results of sensitivity analyses, the difference between beer group and alcohol-free beer group was statistically significant (page 8, lines 172-175).

ANSWER: thank you for this clarification, we deleted this sentence since not supported by data.

---

## [Decision Letter · Decision Letter 2]

11 May 2020

To beer or not to beer: a meta-analysis of the effects of beer consumption on cardiovascular health

PONE-D-20-01229R2

Dear Dr. Cignarelli,

We are pleased to inform you that your manuscript has been judged scientifically suitable for publication and will be formally accepted for publication once it complies with all outstanding technical requirements.

With kind regards,

Yan Li, MD, PhD

Academic Editor

PLOS ONE

Additional Editor Comments (optional):

Reviewers' comments:

Reviewer's Responses to Questions

**Comments to the Author**

1. If the authors have adequately addressed your comments raised in a previous round of review and you feel that this manuscript is now acceptable for publication, you may indicate that here to bypass the “Comments to the Author” section, enter your conflict of interest statement in the “Confidential to Editor” section, and submit your "Accept" recommendation.

Reviewer #1: All comments have been addressed

Reviewer #2: (No Response)

2. Is the manuscript technically sound, and do the data support the conclusions?

Reviewer #1: Yes

Reviewer #2: (No Response)

3. Has the statistical analysis been performed appropriately and rigorously? 

Reviewer #1: Yes

Reviewer #2: (No Response)

4. Have the authors made all data underlying the findings in their manuscript fully available?

Reviewer #1: Yes

Reviewer #2: (No Response)

5. Is the manuscript presented in an intelligible fashion and written in standard English?

Reviewer #1: Yes

Reviewer #2: (No Response)

6. Review Comments to the Author

Reviewer #1: (No Response)

Reviewer #2: (No Response)

7. PLOS authors have the option to publish the peer review history of their article (what does this mean?). If published, this will include your full peer review and any attached files.

Reviewer #1: No

Reviewer #2: No

---

## [Editor Report · Acceptance letter]

13 May 2020

PONE-D-20-01229R2 

To beer or not to beer: a meta-analysis of the effects of beer consumption on cardiovascular health 

Dear Dr. Cignarelli:

I am pleased to inform you that your manuscript has been deemed suitable for publication in PLOS ONE. Congratulations! Your manuscript is now with our production department. 

With kind regards,

on behalf of

Professor Yan Li 

Academic Editor

PLOS ONE